# Nutritional Management of Intestinal Failure due to Short Bowel Syndrome in Children

**DOI:** 10.3390/nu15010062

**Published:** 2022-12-23

**Authors:** Maria Giovanna Puoti, Jutta Köglmeier

**Affiliations:** Unit of Nutrition and Intestinal Failure Rehabilitation, Department of Paediatric Gastroenterology, Great Ormond Street Hospital for Children NHS Foundation Trust, Great Ormond Street, London WC1N 3JH, UK

**Keywords:** short bowel syndrome, intestinal failure, nutritional rehabilitation, enteral feeding, parental nutrition, children, intestinal adaptation

## Abstract

Background: The most common cause of intestinal failure (IF) in childhood remains short bowel syndrome (SBS), where bowel mass is significantly reduced due to a congenital atresia or resection and parenteral nutrition (PN) needed. Home PN has improved outcome and quality of life, but the long-term therapeutic goal is to achieve enteral autonomy whilst avoiding long term complications. This paper is aimed at discussing nutritional strategies available to clinicians caring for these patients. Methods: A literature search was performed from 1992 to 2022 using Pubmed, MEDLINE and Cochrane Database of Systematic Reviews, and recent guidelines were reviewed. In the absence of evidence, recommendations reflect the authors’ expert opinion. Results: Consensus on the best possible way of feeding children with IF-SBS is lacking and practice varies widely between centres. Feeding should commence as soon as possible following surgery. Oral feeding is the preferred route and breast milk (BM) the first milk of choice in infants. Donor BM, standard preterm or term formula are alternatives in the absence of maternal BM. Extensively hydrolysed or amino acid-based feeds are used when these are not tolerated. Solids should be introduced as soon as clinically appropriate. Children are encouraged to eat by mouth and experience different tastes and textures to avoid oral aversion. Aggressive weaning of PN and tube (over-) feeding are now discouraged. Conclusions: To date, uniform agreement on the optimal type of feed, timing of food introduction and feeding regime used is lacking and great difference in practice remains. There is need for more research to establish common treatment protocols.

## 1. Introduction

Intestinal failure (IF) is the reduction in intestinal mass or function below the minimal amount necessary for adequate digestion, absorption and appropriate growth in children [1]. It leads to the inability to maintain nutritional, fluid, and electrolyte status while consuming a regular oral diet. Clinically, children require parental nutrition (PN) for a prolonged period whilst attempts are made to build up enteral and oral feeds [2,3]. The most common cause of IF in children is short bowel syndrome (SBS) [4]. Its incidence is estimated to be 24.5 per 100,000 live births with a higher incidence in preterm infants born before 37 weeks gestation [5]. A report published by the British Society of Parenteral and Enteral Nutrition suggested a 100-fold increase in patients with SBS-IF over a 10-year period between 1992 to 2012 and a similar further rise of numbers during the following decade [6]. Better survival rates and management are the likely explanation.

SBS is a clinical condition characterized by loss of digestive and absorptive gut surface due to extensive surgical resection reducing the bowel length below a critical value for adequate supply of nutrients and/or fluids. In general, it is defined as having <25% of small intestinal length predicted for gestational age or a prolonged period of need for PN (>6 weeks) after bowel resection [7]. Although the cut-off length is related to many factors, a remnant bowel of less than 40 cm of small bowel is associated with occurrence of IF-SBS [8,9].

Causes are heterogenous and vary between younger and older children. In infants it may result from congenital disease such as intestinal atresia, gastroschisis and long segment Hirschsprung disease (total/subtotal intestinal aganglionosis) or post-natal diseases such as necrotizing enterocolitis, mid gut volvulus and vascular thrombosis. In older children it can develop secondary to volvulus, inflammatory bowel disease, trauma and intestinal ischaemia [1,10]. Necrotizing enterocolitis is the most common cause of SBS currently accounting for 26% of all cases [11].

Anatomically, SBS can be divided in three subtypes, which helps to council parents about the likely outcome: type I describes an end-jejunostomy, type II a jejuno-colonic anastomosis and type III a jejuno-ileocolonic anastomosis where ileo-colonic (IC) valve and colon are preserved [12].

The consequences of SBS are malabsorption, diarrhoea, dehydration, electrolyte imbalances, nutrients deficiencies and weight loss or poor weight gain resulting in the need for nutritional support with parental and enteral feeding to maintain growth velocity. In addition to residual small bowel length and anatomical situation, other factors impact on the long-term outcome including gestational age in infants, part of bowel resected, functioning of remaining bowel, underlying diagnosis, long-term presence of stoma vs. presence of primary anastomosis, number of surgical procedures, neuropathy related motility problems particularly in gastroschisis or atresia and age of the patient at the time of surgery [8,9,13,14,15,16,17,18,19]. At birth term, newborns are expected to have a small bowel length of around 250 ± 40 cm. Rapid bowel growth occurs during the last trimester of pregnancy. Preterm infants starting out with the same residual bowel length as a term infant or older child are hence expected to have more potential for gut adaptation.

The consequences of SBS vary ranging from single micronutrient malabsorption to complete IF, depending mainly on the remaining length of the small intestine. The loss of specific parts of the gut determines specific macro-and micronutrient deficiencies such as Vitamin B12 deficiency or bile acid-induced diarrhoea from resection of terminal ileum. Understanding the anatomical situation the child is left with is hence important to tailor the management accordingly. The duodenum is responsible for breakdown of food in the small intestine by using enzymes and regulates the rate of emptying of the stomach via hormonal pathways. In addition, the release of secretin and cholecystokinin from duodenal epithelial cells occurs in response to acidic and fatty stimuli present when the pylorus opens and emits gastric chyme into the duodenum for further digestion. This in turn causes the liver and gall bladder to release bile and the pancreas to release bicarbonate and digestive enzymes such as trypsin, lipase and amylase into the duodenum [20]. The jejunum makes around 40% of small bowel length and has longer villi compared to duodenum and ileum. Sugars, amino- and fatty acids are absorbed here and fructose through passive transport, whilst amino acids, smaller peptides, vitamins and glucose are actively transported. The absorption of vitamin B12 and products of digestion that were not absorbed by the jejunum take place in the ileum as well as reabsorption of bile salts.

Children with a jejunoileal anastomosis, where most of the jejunum has been removed, have a better chance to achieve enteral autonomy, as this proximal resection has spared the ileum, which has the best capacity to adapt both structurally and functionally [21]. In the colon water is absorbed against a transmucosal pressure gradient and also the remaining absorbable nutrients and vitamins created by colonic bacteria including vitamin K, thiamine and riboflavin. Indigistible matter is pushed to the rectum. Potassium is secreted into the lumen and dietary fibres fermented to short chain fatty acids [20]. The cornerstone of SBS management in nutritional rehabilitation and to restore nutritional autonomy. The ultimate goal is to wean PN whilst avoiding long term consequences such as intestinal failure associated liver disease (IFALD), recurrent central venous catheter (CVC) related blood stream infections (CRBSIs), loss of central venous access sites, small bowel bacterial overgrowth, nephrocalcinosis and metabolic bone disease [22]. D-lactic acidosis occurs in patients with short bowel syndrome who have part or all of the colon in continuity. Undigested carbohydrates act as a substrate for bacteria in the colon. The rise in D-lactic acid levels in the blood stream causes neurological symptoms such as confusion, slurring of speech and unsteady gate and is often referred to as D-lactate encephalopathy. Therapeutic strategies include the use of antibiotics to sanitise the abnormal flora of the colon and dietary manipulations to reduce the amount of ingested carbohydrates.

Intestinal transplantation offers an opportunity for intestinal recovery in those who have developed complications without any hope for establishing enteral autonomy in the unforeseeable future [23].

Given the heterogeneity and complexity of this population, intestinal rehabilitation should take place within the set-up of a multidisciplinary IF rehabilitation team staffed with a paediatric gastroenterologist, dietician, pharmacist, clinical nurse specialist and surgeon to deliver holistic care and guarantee the best outcome [18]. Access to interventional radiology (IR) for CVC management or a surgeon trained in IR catheter insertion technique are essential to avoid loss of venous access sites.

A clinical psychologist and other health professionals such as the speech and language therapist are desirable [24,25].

This review aims to summarize the current knowledge available in the literature on nutritional management of children with SBS.

## 2. Materials and Methods

A literature search was performed from 1992 to 2022 using Pubmed, MEDLINE and Cochrane Database of Systematic using the search terms “short bowel syndrome, intestinal failure, nutritional rehabilitation, enteral feeding, parental nutrition, children and intestinal adaptation”. In addition, recent guidelines were reviewed. Literature available on SBS is mostly based on retrospective experience of single centres. In the absence of evidence, recommendations reflect the authors’ expert opinion. A total of 80 papers were considered relevant and referenced in this review to give an overview of the current evidence available.

## 3. Results

### 3.1. Intestinal Adaption

Intestinal adaption after bowel resection is a complex physiological process by which the intestine goes through structural and functional changes to achieve fluid and nutrient absorption in the remnant bowel [26]. It involves bowel lengthening, both in length and diameter resulting in villous hyperplasia, crypt cells proliferation, muscular hypertrophy and angiogenesis [27]. 

The degree of adaption depends on several variables including residual bowel length, type and quality of residual intestine, presence of ileocecal valve and colon, intestinal continuity and nutrition status [28].

Residual bowel length can predict probability and time of weaning from nutritional support. A residual small bowel length of >15 cm with an ileocecal valve or >40 cm without an ileocecal valve is associated with favourable outcome in term infants. The small bowel of preterm infants has higher potential to grow because the elongation of small bowel occurs in the third gestational trimester. Weaning off PN can eventually occur in preterm neonates with >10% of remaining small bowel [29,30].

The type of residual bowel is another important factor impacting on intestinal adaptation. The ileum has greater potential for adaption compared to the jejunum even if the surface area is less. This is because the ileum absorbs most nutrients (bile salts, vitamin B12, fat soluble vitamins, electrolytes) and has a slower motility pattern due to the proximity to the ileocecal valve, which allows a longer mucosal contact and higher fluid and electrolyte uptake [31]. The ileal break controls transit of a meal through the gastrointestinal tract through a primary inhibitory feedback mechanism and therefore enhances nutrient absorption [32]. The presence of the ileocecal valve and at least part of the colon also prevents the retrograde flow of bacteria into the small bowel [18,33].

Use of the gastrointestinal tract promotes intestinal adaptation especially through oral feeding (OF), which is the most physiological route. In general, the luminal nutrients stimulate epithelial cells, secretion of trophic gastrointestinal hormones, pancreatic enzymes and bile. Stimulation of blood flow prevents mucosal atrophy, loss of barrier function and downregulation of the gut immune system [34,35]. It has been postulated that a higher nutrient complexity represents a greater digestive workload which challenges the digestive and absorptive function of the remnant gut more and therefore results in more pronounced hyperplasia—the key to intestinal adaptation [28,36].

Whilst adaptive processes occur, PN should be used to provide appropriate hydration and intake of macro- and micronutrients to guarantee adequate growth until full enteral autonomy can be achieved. Enteral nutrition (EN) or OF should be started as soon as possible after surgery and PN stopped when appropriate nutritional requirements are met with only OF and/or EN.

### 3.2. Nutritional Management

When considering nutritional management of SBS it is important to pay attention to the post-surgical clinical course of SBS and physiological processes according to the time passed since the initial insult and surgical repair. This can be divided into three phases: the acute phase, the adaption phase and the maintenance phase [37]. Successful nutritional rehabilitation depends on the individual management of these stages (Figure 1).

The acute phase occurs immediately after the bowel resection. It is characterized by electrolyte disturbances, significant gastrointestinal losses, fluid shift, metabolic alterations, gastric hypersecretion, dysmotility and post-resection ileus. The gastric hypersecretion is caused by the lack of inhibitory hormones, which are normally released from the terminal ileum [37]. Close monitoring to avoid dehydration, acid-base abnormalities, and electrolyte deficiencies is required.

PN is usually necessary at this stage to restore and maintain fluid balance, electrolytes and acid-base balance and provide nutrition support. Gastric/stool losses should be replaced with fluid and electrolyte solutions, nutritional deficiency corrected with adequate intravenous macronutrients and gastric hypersecretion reduced with acid suppression. At the earliest opportunity EN via a nasogastric tube or OF should be started once the surgical ileus has settled down. Fluid and electrolyte requirements at this stage can quickly change and should be reassessed on a regular basis and adjusted accordingly to avoid imbalances.

The second phase is characterized by fluid-shift stabilization as the remnant part of the bowel attempts to increase fluid and nutrient absorption. In addition, structural and functional changes in the remaining small bowel occur during this phase which increase absorptive surface and slow down bowel transit resulting in adaptive hyperphagia. These changes are stimulated by the presence of nutrients in the gastrointestinal lumen which enhance pancreatic and biliary secretions, and hormones released, if present, by the ileum and colon. Nutritional strategies include advancement of EN/OF by a stepwise increase according to tolerance and reduction in PN in parallel. Cyclical PN through reduction in the daily infusion time can be attempted as long as blood glucose levels are maintained.

This phase can start from 48 h up to 4 weeks after the gut resection and can last from weeks up to 18–24 months [37]. The duration will depend on the prognostic factors discussed above. Home PN should be considered if patients are required to be on PN for 3 months and more as long as the child is stable enough to be managed outside the hospital [38].

Finally, during the maintenance phase weaning from PN support should be aimed for. This is the longest phase and can last for several years. A small subgroup of children will remain PN-dependent, and this phase continues indefinitely unless small bowel transplantation is performed [4]. However, the focus of nutritional management is very much centred around achieving enteral autonomy whilst maintaining adequate weight gain, growth, micronutrient status, hydration and electrolyte balance. When it is not possible to establish full OF a combination of EN and OF or EN alone alongside PN should be considered, but aggressive (over-) feeding with EN avoided [4]. Micronutrients such as vitamins, trace metals and iron can be supplemented via the oral route or given intramuscular or subcutaneous injections.

The cornerstone of nutritional rehabilitation of SBS is to combine PN and EN/OF to stimulate intestinal adaption of the remaining bowel in the best possible way. To date the optimal strategy how to progress with feeding into the gut remains subject to debate. Most centres carrying for these young patients have based their clinical practice on personal experience of members of their team and local guidelines. Evidence to support type and mode of feeding largely consist of retrospective observational studies and small case series. High quality randomized controlled trials are only achievable for big multi-centre studies involving large numbers of children, as children with SBS behave very differently according to the underlying primary insult to the gut and resulting anatomical and functional circumstances.

### 3.3. Parental Nutrition

PN is started soon after the bowel resection to provide adequate fluid and calorie requirements. The aim of PN therapy is to provide the nutritional requirements for normal growth and development, while the bowel undergoes the adaptation necessary for the transition to an enterally based diet.

Total energy expenditure and fluid volume should be carefully calculated at the start and a well-balanced nutrient intake provided to maintain growth. Requirements of protein, carbohydrates and fat vary depending on the age of the child and need to be regularly reviewed as both under- and overfeeding should be avoided. Electrolytes, vitamin and minerals should also be monitored regularly to avoid deficiencies [39]. As these children can have high gastrointestinal losses and/or high energy or nutrient requirements standard PN bags may not be suitable and individualized bags should be made. When advancements in enteral nutrition are successful, PN should be decreased and cycled. The aim of cycling PN is to reduce hyperinsulinemia, with subsequent fat accumulation and liver disease as well as to allow for PN free time during the day in children requiring PN more long term.

A dedicated nutritional team with extensive experience in PN involving a paediatric gastroenterologist, PN pharmacist, dietician and PN clinical specialist nurse should lead the management. The European Society of Paediatric Gastroenterology Hepatology and Nutrition (ESPGHAN) provides comprehensive recommendations on how to start and monitor PN, prevent and manage related complications [40].

PN administration requires insertion of a central venous catheter. Initial venous access should be a peripherally inserted central venous catheter. If the patient requires long term PN a cuffed tunnelled central venous catheter is usually placed. All effort should be made to preserve venous access through avoidance of catheter related infections and thrombosis. This increases the chance of successful long-term PN in case needed and a reduced risk to switch from intestinal failure to “nutritional failure” and indication for intestinal transplanatation [41]. Home PN, where one, preferably two parents/carers are trained to give PN outside the hospital setting, is associated with a better quality of life and reduced sepsis risk and should hence be offered to children who are stable and likely going to need PN for 12 weeks and beyond [38]. Modern IF management and improved PN solutions have resulted in reduced complications including intestinal failure associated liver disease (IFALD), renal and metabolic bone disease [42]. The complications children with SBS can experience are summarised in Figure 2.

### 3.4. Enteral Nutrition

EN is the cornerstone of intestinal rehabilitation in SBS. It should be started in parallel with PN as soon as the patient is stable, usually when the child has gone through the early acute phase. There is some evidence to support that early post-operative feeding is associated with reduced time to tolerating full oral feeds [43].

Oral feeding should be allowed whenever possible as more physiological. However, it may not always be achievable as clinical condition, mechanical ventilation, e.g., in premature infants and intestinal dysmotility may prevent it. Enteral feeding through tube feeding should then be considered. Gestational age at presentation, remaining bowel and phase of SBS should be taken into consideration at the start.

Feeds available are breast milk, whole protein formula, extensively hydrolysed and amino acid-based feeds and those enriched with medium chain triglycerides (MCT).

Recommendations for enteral feeding in children with SBS have been made, but there is no consensus of the optimal strategy [3]. Mode of administration (oral vs. continuous vs. bolus feeding), time of introduction, composition of feeds (polymeric vs. hydrolysed vs. elemental formula) are still subject to debate.

### 3.5. Breast Milk

Whilst consensus on the optimal enteral feed formulation for children with SBS has still not been reached, data support the beneficial effect of human breast milk on gut adaptation [44]. In newborns and infants, human milk is now recommended by most carrying for these young SBS patients. Breast milks contains IgA, leucocytes and nucleotides to support the infant’s immune system. Glutamine, growth hormone and epidermal growth factor support bowel adaptation [45,46]. Immunoglobulins and antimicrobial peptides may promote intestinal colonization with appropriate lactobacilli and related bacteria which are important elements of a healthy microbiome [47]. The composition of human milk is complex and influenced by gestation of the infant at birth and postnatal age. It changes dynamically over the lactation period. The protein decreases during the first month of lactation and the main source of carbohydrates are lactose and complex oligosaccharides. Some animal studies suggested that bovine colostrum is beneficial to bowel adaptation, but human studies could not confirm this beneficial effect [48,49,50,51].

The human milk oligosaccharides (HMOs) stimulate enterocyte growth, act as prebiotics and positively influence the human intestinal microbiome [52,53]. The most abundant HMO is 2′-fucosyllactose (2′-FL), which has been shown to reduce intestinal inflammation. In a mouse model 2′-FL significantly reduced the severity of colitis in interleukin-10 null mice by enhancing the integrity of the epithelial layer and shifting the gut microbiota to a more positive environment [53]. Furthermore 2′-FL could improve intestinal adaptation after resection of the ileo-caecum through enhancement of energy usage by the intestinal microbiome [52,53,54]. A retrospective review of 30 infants with SBS demonstrated that the use of breast milk was significantly associated with shorter duration of PN. However, due to the small number of patients and possible selection bias definitive conclusions could not be made [55].

Donor breast milk offers an alternative form of enteral nutrition for preterm or low birth weight (LBW) infants in the absence of maternal breast milk. Donor breast milk may retain some of the non-nutritive benefits of maternal breast milk, but uncertainty exists about the balance of risks and benefits of feeding formula versus donor breast milk for preterm or LBW infants [56]. As comparison between maternal and donor breast milk in SBS has not been evaluated recommendations about the use of donor milk can hence not be made.

When not enough breast milk is available feeds can be combined with a whole protein formula to reach requirements, as protein digestion and absorption is completed in the upper small intestine and whole protein should generally not cause a significant problem in SBS.

### 3.6. Specialised Formula Feeds

If the child cannot tolerate whole protein milk, either because there is insufficient luminal surface area for digestion and absorption or due to cow’s milk protein allergy, an extensively hydrolysed feed (EHF) can be offered instead. Many authors consider EHFs the first formula of choice if breast milk is not available. Amino acid-based feeds (AAF) are tried when an EHF has also failed. What intestinal adaptation is concerned, there appears to be no difference between the use of aminoacid and hydrolysed feeds. Whilst complex proteins may be superior in stimulating adaptation, infants with SBS can have increased bowel permeability and are thus at risk of developing food allergies. Although no convincing data to support this hypothesis exist, an association between non-infectious eosinophilic colitis and SBS has been described [57,58]. A small case series of four children with whole protein intolerance offered EHF who required PN for 6 months or more and were unable to make progress with PN weaning, showed improved enteral tolerance and decreased gut permeability when feeds were changed to AAF. Another case review of four patients with SBS found that two of the children tolerated feeds better when changed from EHF to AAF [59,60]. A retrospective review demonstrated a shorter duration of PN in 30 infants fed with an AAF [55].

Should even AAF intolerance exist, a modular feed (MF) approach can be tried, as long as a dietitian with expertise in MF and a feed production unit are available [61].

### 3.7. Route of Feeding

Enteral feeds can be administered orally, via naso-gastric or gastrostomy as bolus or continuous feeds and the jejunal route in upper GI dysmotility. Breast feeding, or if not feasible bottle feeding, should be started in small volumes as soon as possible in neonates to stimulate suck and swallow reflexes. Solid foods may be introduced at the age of 4–6 months to stimulate oral motor activity and to avoid feeding aversion behaviour. Gestational age at birth should however be taken into account in premature infants, who may not be mature enough for complementary feeding [62,63].

Both bolus and continuous feeding have benefits and drawbacks. Bolus feeds are more physiological and promote bile flow. Continuous feeds are thought to improve absorption by reducing gastrointestinal transit and maximizing mucosal contact with nutrients whilst reducing the risk of osmotic diarrhoea. However, as fasting periods are suppressed when continuous feeding is offered, gut motility can be negatively affected which plays a role in the development of small bowel bacterial overgrowth [64]. Bolus feeding on the other hand allows alternation between fasting and feeding periods and can hence improve gut motility. In addition, it stimulates the release of trophic hormones which promote intestinal adaption. Furthermore, feeding breaks avoid permanent secretion of insulin with consequent fat synthesis and deposition with the risk of liver steatosis and increased fat body mass.

Continuous feeding can however increase feed tolerance when bolus feeds are not tolerated.

Feeds are best given gastrically, and the jejunal route should be reserved for those with severe foregut dysmotility as it can further reduce the small bowel mass available for absorption, increase the risk of diarrhoea and limit the chance to transition to oral feeding. Continuous administration of a prolonged period is needed due to the limit of volume which can be given directly into the small bowel. Children are hence connected to a feeding pump for long a period of time, which reduces the chance to engage in normal age-appropriate childhood activities and increases the burden of care for the parents. Therefore, jejunal feeding should only be considered for patients with severe gastroesophageal reflux (GERD) with risk of aspiration and gastric dysmotility.

### 3.8. Solids

Intraluminal nutrients have a stimulating effect on the gut epithelial cells and the production of trophic hormones. They increase pancreatic and biliary secretion and prevent mucosal atrophy, loss of barrier function and downregulation of the mucosal immune system [65,66].

Stimulation of mucosal hyperplasia occurs through direct contact with epithelial cells through secretion of trophic gastrointestinal hormone secretion [35].

The more digestion a nutrient (e.g., whole protein) needs, the more hyperplasia it will cause. In other words, it is key to gut adaptation. The composition of the diet should balance gastrointestinal tolerance with specific nutrients in a complex form to stimulate the adaptive process further [36,67].

Whole proteins are preferred in terms of increased workload to the digestive and absorptive function of the bowel.

Glutamine is the main fuel for enterocytes and thought to enhance mucosal hyperplasia. 

However, glutamine supplementation in infants with gastrointestinal disease showed no difference in PN dependence, feeding tolerance, and intestinal absorptive-barrier function compared to placebo [68,69].

Disaccharides are more trophic than monosaccharides due to the higher functional workload of the bowel. Lactose intolerance may occur in patients with proximal jejunum resection [36].

A cross-over study in adults with SBS, however, demonstrated a similar tolerance of a lactose-free diet compared to a diet containing 20 g of lactose a day. Lactose may promote the production of short chain fatty acids (SFCAs) in the colon and a small amount of lactose can therefore be offered to infants [4,70].

Insoluble forms of fibre, such as cellulose found in cereals bind to water and contribute to bulking and softening of the stool. Soluble forms such as pectin and guar gum found in fruit and vegetables slow gastric emptying and the overall gut transit time and have a mild anti-diarrhoeal effect [71].

Soluble fibre and some starches pass into the colon undigested, where they are fermented by colonic bacteria into SFCA’s, which can account for 5–10% of the energy intake. Starch is the main carbohydrate substrate for colonic bacterial fermentation in patients with SBS.

Pectin also increases SFCA production and improves fluid absorption. The SCFA Butyrate has trophic effects on the cells in the jejunum and ileum when delivered to the colon [72,73]. 

The absorption of long chain triglycerides (LCTs) is dependent on bile. They enhance bowel adaptation and stimulate the secretion of PYY and glucagon-like-peptide 2 (GLP-2) which mediates the ileal and jejunal brake and hence slows down transit time.

Furthermore, LCTS have anti-inflammatory properties and improve the splanchnic circulation [74,75].

Medium chain triglycerides (MCTs) are directly absorbed across the enterocyte into the portal circulation. This process already starts in the stomach and increases the uptake of fat. Dense energy is hence available faster. However, a high MCT diet can cause osmotic diarrhoea as a result of rapid hydrolysis of MCTs in patients with jejuno-or ileostomy, and caution is advised [76]. In patients with an intact colon the ingestion of MCTs improved fat absorption and might be beneficial in patients with bile acid or pancreatic insufficiency. Of note is that MCTs are saturated fats and do not contain essential fatty acids.

Non-nutrient nutritional factors include a variety of nutrients, secretions, and other essential components in the diet or produced in the lumen of the gastrointestinal tract that have been shown to stimulate gut mucosal growth. GLP-2 is produced by the enteroendocrine cells of the terminal ileum and colon in response to luminal nutrients, especially LCTs and carbohydrates [77].

GLP-2 increases the mucosal surface area of the gut and increases nutrient absorption. It helps to slow down gut motility, improves gut-barrier function and increases blood flow to the gut.

Synthetic forms are now available and licensed for children from the age of one who have SBS and fail to make progress with weaning from PN.

### 3.9. Microbiome

The human gastrointestinal tract contains around 10^14^ microorganisms including bacteria which are commonly referred to as the gut microbiome. The understanding of the human microbiome has greatly increased both in health and disease in recent years. Patients with IF were found to have an overall reduction in bacterial diversity with an increase in Proteobacteria, Enterobacteriaceae, Lactobacilli and a decrease in Bacteroidetes [78]. 

Microbiome analysis could hence potentially be used as biomarker to guide clinical teams trying to stimulate intestinal adaptation. Modification of the microbiome could offer a new therapeutic approach to increase enteral tolerance and absorption.

A balanced diet can contribute to a healthy microbiome.

### 3.10. Blended Diet

A blended diet (BD), where blended foods are administered into an enteral feeding tube, can be considered in children with severe oral aversion who refuse solid food or those with swallowing dysfunction. There is evolving evidence to suggest that a BD can improve feeding difficulties, GERD and bowel function. In a small study of ten paediatric intestinal failure patients who were more than one year of age, successful transition from an elemental formula to a formula with real food ingredients could be achieved [79].

BD can increase bacterial diversity and species richness and therefore has a favourable influence on the gut microbiome.

However, more studies are needed to provide evidence that a blended diet is a safe and effective feeding strategy in children with IF-SBS [80].

## 4. Conclusions

The nutritional management of infants and children with IF-SBS aims to promote gut adaptation to allow normal growth and development. Most children require a period of parenteral nutrition and modern IF rehabilitation should aim to support intestinal adaptation whilst avoiding complications. The ultimate goal is to achieve intestinal autonomy and therefore weaning from PN. To date high powered trials to reach a uniform consensus on the best type of feed, timing of food introduction and feeding regime used are lacking and great difference in practice remains.

There is need for large multi-centre trials to establish common treatment protocols.

## Figures and Tables

**Figure 1 nutrients-15-00062-f001:**
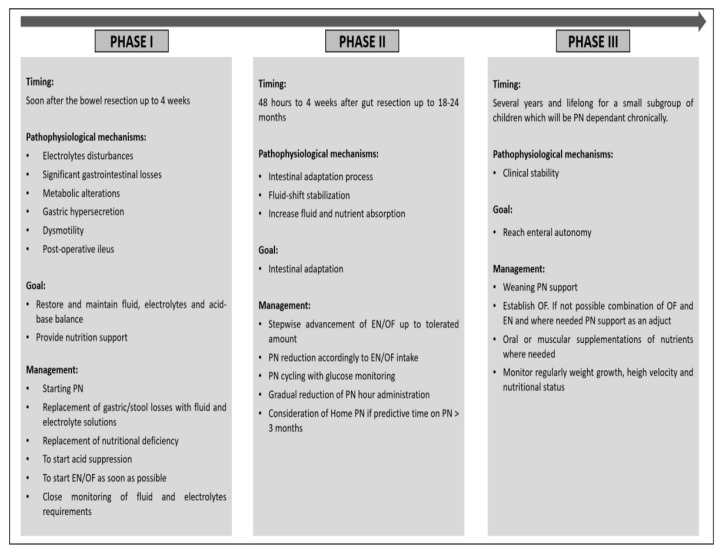
Nutritional management accordingly to pathophysiological phases of SBS.

**Figure 2 nutrients-15-00062-f002:**
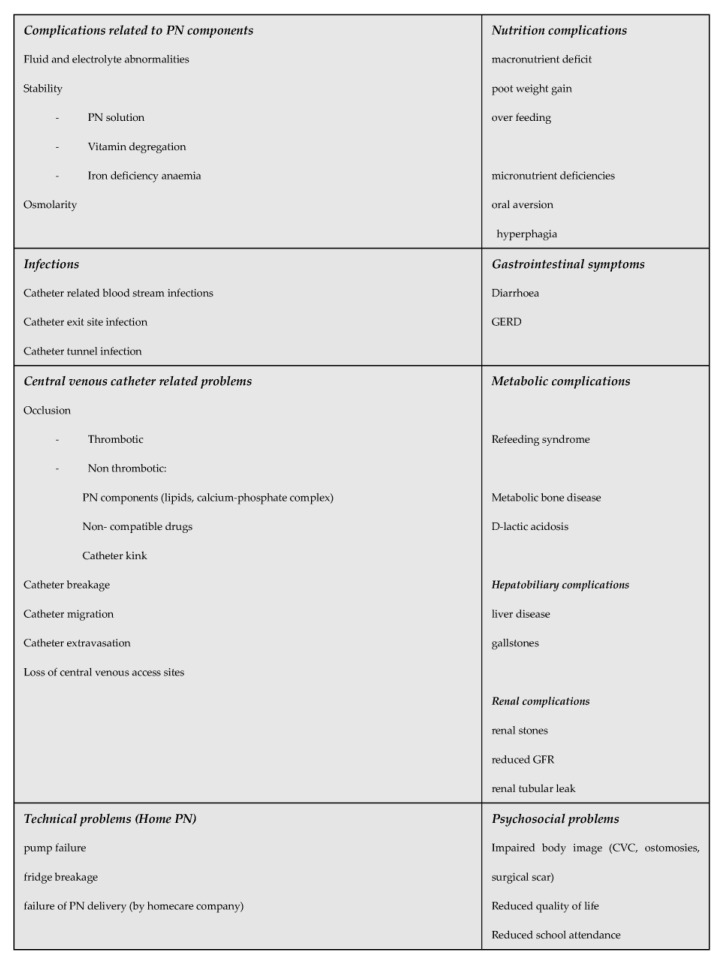
Complications of children with SBS receiving long term parenteral nutrition.

## Data Availability

A literature search was performed from 1992 to 2022 using Pubmed, MEDLINE and Cochrane Database of Systematic Reviews and recent guidelines reviewed. In the absence of evidence, recommendations reflect the authors’ expert opinion.

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
