# Peer review of "Nutritional Management of Intestinal Failure due to Short Bowel Syndrome in Children"

_nutrients, 2022, doi:10.3390/nu15010062_

Round 1

Reviewer 1 Report

Dear authors, congratulations on this noteworthy contribution. The current study's design is reasonable, and the topic is interesting to the field. However, it would be necessary to review some methodological aspects before continue.

# Should the title of the manuscript include the field to which it is addressed? (Nutritional management of intestinal failure due to short bowel syndrome in children?)

# I think section 2, "Material and methods" (lines 116-120), is too succinct. It is not clear whether it is a systematic review or a narrative, or an expert review. I think the authors should be more explicit in describing the methodology followed to obtain the information by the sigma statement: an evidence-based guide consisting of a checklist and flowchart intended to be used as tools for authors seeking to write systematic literature reviews (SLR) and meta-analyses (PMID: 29283007 & PMID: 26830668).

Some clinical consequences of short bowel syndrome are described between lines 100 and 105. Perhaps mention should be made of D-the -lactic acidosis, also referred to as D-lactate encephalopathy. This rare neurologic syndrome occurs in individuals with short bowel syndrome or following jejuno-ileal bypass surgery. Symptoms typically present after ingesting high-carbohydrate feedings (PMID: 16306301; PMID: 27522622).

# The letter size in Figure 1 is too small and is not reader-friendly. 

# The manuscript lacks figures related to the topic and some tables with complementary information (e.g., side effects and/or contraindications to using enteral or parenteral nutrition or perhaps the causes and consequences of the refeeding syndrome).

Once again, our congratulations to the authors for this input. 

  •  

Author Response

Point by point response to the comments of reviewer 1:
# Should the title of the manuscript include the field to which it is addressed? (Nutritional management of intestinal failure due to short bowel syndrome in children?)
Response
We agree with the reviewers comment and have added ‘in children’ to the title of the manuscript (line 3)
# I think section 2, "Material and methods" (lines 115-123), is too succinct. It is not clear whether it is a systematic review or a narrative, or an expert review. I think the authors should be more explicit in describing the methodology followed to obtain the information by the sigma statement: an evidence-based guide consisting of a checklist and flowchart intended to be used as tools for authors seeking to write systematic literature reviews (SLR) and meta-analyses (PMID: 29283007 & PMID: 26830668).
Response
This invited review is not aimed to be a systematic review or meta-analysis. It gives an overwiew of the current available literature. A literature search was performed using Pubmed, MEDLINE and Cochrane Database of Systematic Reviews from 1992 to 2022 using the search terms ‘short bowel syndrome, intestinal failure, nutritional rehabilitation, enteral feeding, parenteral nutrition, intestinal adaptation and children’. In addition, recent guidelines were reviewed. Literature available on SBS is mostly based on retrospective experience of single centres. In the absence of evidence, recommendations reflect the authors’ expert opinion. A total of 80 papers were considered relevant to the nutritional management of intestinal failure due to short bowel syndrome in children and referenced in this review to give an overview of the current evidence available in the literature. Section 2 has been modified accordingly (lines 115 – 123).
#  Some clinical consequences of short bowel syndrome are described between lines 100 and 105. Perhaps mention should be made of D-the -lactic acidosis, also referred to as D-lactate encephalopathy. This rare neurologic syndrome occurs in individuals with short bowel syndrome or following jejuno-ileal bypass surgery. Symptoms typically present after ingesting high-carbohydrate feedings (PMID: 16306301; PMID: 27522622).
Response
 A paragraph on D-lactic acidosis has been included in the list of potential complications of SBS   in the revised manuscript (lines 102 – 108)
# The letter size in Figure 1 is too small and is not reader-friendly. 
Response
The font size in figure 1 has been increased 
# The manuscript lacks figures related to the topic and some tables with complementary information (e.g., side effects and/or contraindications to using enteral or parenteral nutrition or perhaps the causes and consequences of the refeeding syndrome).
Response
Additional information has been included into the manuscript as follows:
Figure 2: Complications of children with SBS receiving long term parenteral nutrition (line 110 – 112)

Reviewer 2 Report

Dear authors Your manuscript entitled "Nutritional management of intestinal failure due to short bowel syndrome" is very interesting and innovative. It raises a very important topic of intestinal failure in the case of short bowel syndrome. If proper nutritional treatment is not introduced, short bowel syndrome is a life-threatening condition due to chronic malnutrition. It should be remembered that this is the result and the result of other diseases of the abdominal cavity, which are almost always associated with the patient's stay in the hospital. Since the symptoms of short bowel syndrome occur during hospitalization, the diagnosis is made by the doctor in charge of the patient in the ward, and the patient leaving the hospital should already be educated about his disease and nutrition. He should also be under the care of a specialist outpatient clinic for the treatment of parenteral and enteral nutrition. Such clinics are located in virtually every major city in Poland, and their addresses are available in hospitals and general practitioners. The doctor establishes the diagnosis based on information about the underlying disease and/or the extent of the small and large intestine resection. In addition, short bowel syndrome is indicated by the occurrence of profuse, debilitating diarrhea in the postoperative period, leading to water and electrolyte disorders, which correlate with the extent of the resection of the gastrointestinal tract. It may also be helpful to perform additional laboratory tests from the collected blood and urine sample, such as complete blood count, biochemistry, including the concentration of important microelements (e.g. magnesium, zinc, selenium), general urine test and 24-hour urine collection, which will allow to observe deficiencies nutrients related to absorption disorders. It should be noted, however, that the basic parameter is the clinical symptoms (diarrhea), the general condition of the patient and the progressive symptoms of dehydration and cachexia. In order to prevent nutritional deficiencies in the short bowel syndrome, parenteral nutrition is introduced as early as possible to prevent malnutrition and weight loss. It consists in the fact that all the necessary nutrients are introduced into the body through the intravenous route. For this purpose, access to a central vein, peripheral veins (usually in the forearm), as well as an arteriovenous fistula (in patients in whom another catheter cannot be inserted) can be used. In addition to parenteral nutrition, enteral nutrition (through a feeding tube, gastrostomy, jejunostomy or microjejunostomy) should be carried out in parallel in order to accelerate the adaptation changes of the intestine. The amount of enteral nutrition administered depends on the amount of excreted stool, but should not exceed 30-40 ml/kg bw/day. In addition, many patients can ingest small amounts of food orally, which often allows them to avoid enteral nutrition. In the case of enteral nutrition, if the administration of a larger amount of food at one time causes diarrhea, the patient should be given the diet continuously in an infusion (e.g. at night). Due to many factors, as well as disease entities, as a result of which short bowel syndrome may develop, it is impossible to clearly define the procedure that would protect against the disease.

I believe that the manuscript "Nutritional management of intestinal failure due to short bowel syndrome" is extremely important for both patients and medical staff. The actions described in it may be an important factor increasing the patient's comfort and improving his condition during hospitalization. It has a very practical application, and the knowledge contained in it should be widely disseminated in medical environments.

Author Response

Point by point response to the comments of reviewer 2
Dear authors Your manuscript entitled "Nutritional management of intestinal failure due to short bowel syndrome" is very interesting and innovative. It raises a very important topic of intestinal failure in the case of short bowel syndrome. If proper nutritional treatment is not introduced, short bowel syndrome is a life-threatening condition due to chronic malnutrition. It should be remembered that this is the result and the result of other diseases of the abdominal cavity, which are almost always associated with the patient's stay in the hospital. Since the symptoms of short bowel syndrome occur during hospitalization, the diagnosis is made by the doctor in charge of the patient in the ward, and the patient leaving the hospital should already be educated about his disease and nutrition. He should also be under the care of a specialist outpatient clinic for the treatment of parenteral and enteral nutrition. Such clinics are located in virtually every major city in Poland, and their addresses are available in hospitals and general practitioners. The doctor establishes the diagnosis based on information about the underlying disease and/or the extent of the small and large intestine resection. In addition, short bowel syndrome is indicated by the occurrence of profuse, debilitating diarrhea in the postoperative period, leading to water and electrolyte disorders, which correlate with the extent of the resection of the gastrointestinal tract. It may also be helpful to perform additional laboratory tests from the collected blood and urine sample, such as complete blood count, biochemistry, including the concentration of important microelements (e.g. magnesium, zinc, selenium), general urine test and 24-hour urine collection, which will allow to observe deficiencies nutrients related to absorption disorders. It should be noted, however, that the basic parameter is the clinical symptoms (diarrhea), the general condition of the patient and the progressive symptoms of dehydration and cachexia. In order to prevent nutritional deficiencies in the short bowel syndrome, parenteral nutrition is introduced as early as possible to prevent malnutrition and weight loss. It consists in the fact that all the necessary nutrients are introduced into the body through the intravenous route. For this purpose, access to a central vein, peripheral veins (usually in the forearm), as well as an arteriovenous fistula (in patients in whom another catheter cannot be inserted) can be used. In addition to parenteral nutrition, enteral nutrition (through a feeding tube, gastrostomy, jejunostomy or microjejunostomy) should be carried out in parallel in order to accelerate the adaptation changes of the intestine. The amount of enteral nutrition administered depends on the amount of excreted stool,but should not exceed 30-40 ml/kg bw/day. In addition, many patients can ingest small amounts of food orally, which often allows them to avoid enteral nutrition. In the case of enteral nutrition, if the administration of a larger amount of food at one time causes diarrhea, the patient should be given the diet continuously in an infusion (eg. at night). Due to many factors, as well as disease entities, as a result of which short bowel syndrome may develop, it is impossible to clearly define the procedure that would protect against the disease.I believe that the manuscript "Nutritional management of intestinal failure due to short bowel syndrome" is extremely important for both patients and medical staff. The actions described in it may be an important factor increasing the patient's comfort and improving his condition during hospitalization. It has a very practical application, and the knowledge contained in it should be widely disseminated in medical environments. 
 Response
We thank reviewer 2 for the compliments. We cannot see from the statement above that the reviewer would like us to make any changes to the manuscript. The reviewer simply gives a short summary of short bowel syndrome, which is extensively discussed in our manuscript. We have only made one small adjustement to the following sentence and added’ where one, preferably two parents/carers are trained to give PN outside the hospital setting, to explain the process of home PN better:
‘Home PN, where one, preferably two parents/carers are trained to give PN outside the hospital setting, is associated with a better quality of life and reduced sepsis risk and should hence be offered to children who are stable and likely going to need PN for 12 weeks and beyond’ (page line 256 – 257)